# Stakeholder-led understanding of the implementation of digital technologies within heart disease diagnosis: a qualitative study protocol

Kamilla Abdullayev ,[1] Timothy JA Chico,[2] Matthew Manktelow,[3] Oliver Buckley,[4] Joan Condell,[3] Richard J Van Arkel,[5] Vanessa Diaz,[6,7] Faith Matcham[1]

For numbered affiliations see end of article.

**Correspondence to**
Kamilla Abdullayev;
kga21@sussex.ac.uk

## ABSTRACT

**Introduction** Cardiovascular diseases are highly prevalent among the UK population, and the quality of care is being reduced due to accessibility and resource issues. Increased implementation of digital technologies into the cardiovascular care pathway has enormous potential to lighten the load on the National Health Service (NHS), however, it is not possible to adopt this shift without embedding the perspectives of service users and clinicians.

**Methods and analysis** A series of qualitative studies will be carried out with the aim of developing a stakeholder-led perspective on the implementation of digital technologies to improve holistic diagnosis of heart disease. This will be a decentralised study with all data collection being carried out online with a nationwide cohort. Four focus groups, each with 5–6 participants, will be carried out with people with lived experience of heart disease, and 10 one-to-one interviews will be carried out with clinicians with experience of diagnosing heart diseases. The data will be analysed using an inductive thematic analysis approach.

**Ethics and dissemination** This study received ethical approval from the Sciences and Technology Cross Research Council at the University of Sussex (reference ER/FM409/1). Participants will be required to provide informed consent via a Qualtrics survey before being accepted into the online interview or focus group. The findings will be disseminated through conference presentations, peer-reviewed publications and to the study participants.

## STRENGTHS AND LIMITATIONS OF THIS STUDY

⇒ The study materials have been informed by patient advisory boards, meaning they are sensitive to the experiences of the participants and the clinicians that will be recruited.
⇒ The study will allow an in-depth understanding of the attitudes and experience of people with lived experience of heart disease and clinicians with experience of diagnosing heart disease.
⇒ The use of an online research platform for participant recruitment will disadvantage certain demographic and clinical groups who are less comfortable using online resources.
⇒ The use of thematic analysis will not be free from the influence of the researcher's personal experience and knowledge.

and both physical[10–12] and psychological[13–16] comorbidities.

Although CVDs are highly prevalent and potentially very damaging, they can become considerably more manageable, or even preventable, if diagnosed early enough.[17] Therefore, implementing secondary prevention measures, in the form of early diagnosis and interventions, is the most effective way of reducing the life-threatening or long-term impacts of CVDs on patient health and well-being.[18] Technological advancements within artificial intelligence and data mining may provide the tools for more accurate and effective diagnosis,[19 20] and can contribute to reducing cardiovascular mortality.[21 22]

In addition to facilitating earlier and more accurate detection of CVDs through artificial intelligence, the increased use of 'wearable' ambulatory assessment technology within healthcare has provided solutions for groups facing barriers that are preventing access to primary care, such as transportation limitations in remote locations,[23 24] or reduced mobility or capacity due to mental

## INTRODUCTION

Cardiovascular diseases (CVDs) are the leading cause of global deaths and are highly prevalent across the world.[1] The situation in the UK is no different, with the British Heart Foundation reporting 7.6 million people living in the UK with a heart and circulatory disease in August 2022, and approximately 460 deaths a day.[2] In addition to debilitating cardiovascular symptoms, individuals suffering from CVDs often experience a diminished quality of life,[3–5] financial burdens from medication and treatments,[6–9]

illness.[25–27] As a result, those who would have missed the opportunity to receive a timely diagnosis and appropriate treatment, are no longer being excluded due to logistical limitations. Overall, it is clear that the use of remote monitoring technology in heart disease can increase the likelihood of survival and decrease the burden of CVD on the individual and ultimately on healthcare services.[28 29]

A variety of digital tools that allow more remote healthcare are already integrated into the NHS. For example, patient records can be accessed remotely thanks to use electronic databases; patients' heart rhythms can be monitored over 24 hours from their homes using Holter monitors; and many appointments are carried out on videoconferencing software or telephone calls. Although there is common complaint regarding the quality and modernity of technology being used within the NHS, it is evident—particularly since the COVID-19 pandemic[30 31]—that there has been a rapid increase in the adoption of digital technology within both the NHS[32] and across the world.[33] Accordingly, there has also been increased research activity investigating the use of digital technologies in cardiovascular health.[34]

When examining the growing use of digital health technologies, it is important to highlight the distinction between the types of technologies that are being implemented, specifically between medical grade devices and consumer technologies. Medical-grade devices require a specific regulatory approval to be used and can come in many forms, including implanted defibrillators and pacemakers, and 24-hour ECG. The data collected by these devices is usually physiological and requires specialists to monitor and interpret. On the other hand, consumer technologies can include health apps on our smartphones, which track our sleep, step-count and medication adherence. These types of technologies are usually in the control of the individual, allowing self-monitoring of symptoms and self-management of many conditions, meaning there is less dependence on an already-overwhelmed healthcare system. Therefore, when the increased accuracy of data produced by medical-grade devices is combined with the detailed and personalised information collected by lifestyle technologies, it allows for a more holistic account of patient health to facilitate an accurate and efficient diagnosis of potential heart diseases.

Nevertheless, along with the potential for improving the accuracy and efficiency of diagnosis, the implementation of digital technologies into healthcare poses challenges with usability, feasibility and accessibility for patients,[35–37] particularly those from under-represented groups in society.[38 39] Too often, devices and systems are created without enough information regarding the patients' and clinicians' needs, which ultimately results in poor engagement and low cost-effectiveness.[35] This highlights the importance of using the voices of stakeholders to inform the development and dissemination of digital health tools into cardiovascular care, as well as any other field within

healthcare, otherwise the likelihood of positive effects is very low.

Therefore, a key factor in the growing shift towards the use of digital health tools within any care pathway is to gain a deeper understanding into how the new technology will fit into the existing systems, and to consider the barriers or facilitators to positive engagement from both patients and clinicians.

### Study objectives
The aim of this project is to develop a stakeholder-led understanding of the implementation of digital technologies to improve holistic diagnosis of heart disease. We have set the following three objectives which will be met via a series of qualitative studies with clinicians and individuals with lived experience of heart disease:
1. Develop a deeper understanding of stakeholder experiences of heart disease diagnosis.
2. Understand stakeholder perspectives on the use of digital health tools within the heart disease diagnosis process.
3. Explore the most meaningful and useful way of relaying digital data back to patients and clinicians.

## METHODS AND ANALYSES
### Patient and public involvement
Prior to the starting participant recruitment, this research was reviewed by a cardiovascular patient advisory group based in Sheffield, which involved all participant-facing documents, including the recruitment materials and focus group schedules. This means we can be confident that the language we will use in the focus groups will be accessible and easy to understand, as well as ensuring that we are covering the important aspects of patient experience during diagnosis and when being asked to use digital health tools by their healthcare providers.

We also met with the NIHR Maudsley Biomedical Research Centre's Race, Ethnicity and Diversity advisory group to discuss the implementation of digital technologies into heart disease diagnosis from a cultural and ethnicity perspective. This meeting highlighted the importance of considering how cultural and religious attitudes would affect engagement with digital technologies aiming to collect health data, as well as emphasising the role of family in individuals' healthcare among ethnic minorities. Given the potential exclusion of certain populations because of our recruitment method, this insight will provide a deeper understanding of additional factors that might contribute to the acceptance and engagement of digital technologies aimed to improve holistic diagnosis of heart disease.

### Study design
This will be a qualitative study, which will use both focus group and interview designs. The topic guides were developed based on the study objectives, with an even split between (1) patient/clinician experiences of diagnosis

and (2) perspectives on the use of digital technologies within healthcare.

## Study population

Based on the available time for data collection against the wider project deadlines, we plan to conduct four focus groups with individuals with lived experience of heart disease. We aim to recruit 5–6 individuals per focus group to ensure there is enough time for each participant to share their views and experiences. The following inclusion/exclusion criteria will be applied: Inclusion criteria: Lifetime diagnosis of heart disease (including but not limited to: angina, heart failure, valve disease, abnormal heart rhythms); aged 18 or over; able to speak English at a level sufficient for participation; able to give informed consent for participation. Exclusion criteria: Major cognitive impairment or dementia preventing participation.

Based on the research team's previous experience of conducting qualitative research with clinicians, a pragmatic decision has been made to aim for a target of 10 clinicians who have had any experience (past or present) of diagnosing people with heart disease to be enrolled into interviews. The eligibility criteria for inclusion are at least 6 months of experience in working with heart disease patients in any capacity (including cardiology, nursing, primary care or other healthcare professional in multidisciplinary teams); aged 18 or over; able to speak fluent English; and able to give informed consent for participation.

## Procedure

As we are interested in hearing from participants across a range of disease durations (ie, those with very early symptoms and those who are on established management plans), we plan to recruit via: social media platforms such as LinkedIn, Twitter and Instagram; Prolific; and existing cohorts of people from the investigators' previous research studies who have consented to be contacted for research purposes. Clinicians will be recruited using purposive sampling via personal and professional connections, and social media platforms such as LinkedIn, Twitter and Instagram.

All focus groups and interviews will be carried out online over Zoom, and consent and baseline demographic data will also be collected via online Qualtrics surveys prior to the online study. We expect focus groups to take about 2 hours and interviews to take about 1 hour. The focus groups and interviews will be semistructured and will follow a preapproved question schedule, split into two sections—patient experience of heart disease and views on digital technologies within healthcare (see online supplemental appendix 1). The same researcher will facilitate all the sessions.

Recruitment and data collection began after ethical approval on 14 November 2022 and will continue until the end of March 2023.

## Data analysis plan

Descriptive statistics for demographics, current mental distress levels, confidence using technologies and participant type-specific questions (including length of time in clinical role for clinicians and details on health condition for participants with lived-experience of heart disease) will be presented.

Transcriptions from both focus group and interview recordings will be validated by the team of researchers and coded and analysed using the NVivo software. In line with Braun and Clarke's 2006 recommendations,[40] an inductive thematic analysis approach will be taken, whereby the data from the transcripts will decide the themes, instead of basing them on any previous theoretical basis.

## Ethics and dissemination

This study was reviewed and approved by the Sciences & Technology Cross-School Research Ethics Council at the University of Sussex (reference ER/FM409/1) on 14 November 2022. We intend to write the resulting paper according to the Consolidated Criteria for Reporting Qualitative research guidelines.[41]

**Author affiliations**
[1]School of Psychology, University of Sussex, Brighton, UK
[2]Department of Infection, Immunity and Cardiovascular Disease, The Medical School, The University of Sheffield, Sheffield, UK
[3]School of Computing, Engineering and Intelligent Systems, University of Ulster at Magee, Londonderry, UK
[4]School of Computing Sciences, University of East Anglia, Norwich, UK
[5]Department of Mechanical Engineering, Imperial College London, London, UK
[6]Department of Mechanical Engineering, University College London, London, UK
[7]Wellcome/EPSRC Centre for Interventional and Surgical Sciences, University College London, London, UK

**Acknowledgements** We would like to thank the two patient and public involvement groups that helped to inform the design of this study: The NIHR Maudsley Biomedical Research Centre's Race, Ethnicity and Diversity (READ) advisory group and the Sheffield-based Cardiology Patient group.

**Contributors** Conceptualisation: FM, TC, JC, OB, VD and RVA. Methodology: FM and TC. Investigation: KA and MM. Writing—original draft: KA. Writing—review and editing: FM and TC. Supervision: FM and TC. Project Administration: KA and FM. Funding Acquisition: FM, TC, JC, OB, VD and RVA.

**Funding** This work is supported by the UK Engineering and Physical Sciences Research Council (EPSRC) (grant number: EP/X000257/1).

**Competing interests** None declared.

**Patient and public involvement** Patients and/or the public were involved in the design, or conduct, or reporting, or dissemination plans of this research. Refer to the Methods section for further details.

**Patient consent for publication** Not applicable.

**Provenance and peer review** Not commissioned; externally peer reviewed.

**ORCID iD**
Kamilla Abdullayev http://orcid.org/0000-0001-6233-5955

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
