## [Reviewer comments · BMJ Open]

ARTICLE DETAILS

TITLE (PROVISIONAL)	A stakeholder-led understanding of the implementation of digital technologies within heart disease diagnosis: a qualitative study protocol
AUTHORS	Abdullayev, Kamilla; Chico, Timothy; Manktelow, Matthew; Buckley, Oliver; Condell, Joan; Van Arkel, Richard; Diaz, Vanessa; Matcham, Faith

VERSION 1 – REVIEW

REVIEWER	Ruth Chambers NHS Stoke on Trent
REVIEW RETURNED	07-Apr-2023

GENERAL COMMENTS	The study protocol is very clearly written with references capturing international research underpinning the protocol. The missing element in the text of the protocol (objectives, methods, study population) seems to be lack of recognition of digital exclusion eg the accessibility/availability of the focus group patients having the right technology/digital skills to participate in an online focus group via zoom; and usage of digital tools relating to heart disease clinical support.
---

REVIEWER	Palmira Bernocchi ICS Maugeri SpA SB, Telemedicine Service
REVIEW RETURNED	18-Apr-2023

GENERAL COMMENTS	This study is very interesting and the goals are well designed. In fact, it is essential to have qualitative feedback from patients and clinicians on the use of technology at various levels and on their propensity. Some remarks: The date of approval by the Ethics Committee is not disclosed. In addition, the protocol should indicate whether the study is already ongoing or planned, thus reporting specific dates. Please specify if COREQ reporting guidelines were followed I have a strong doubt about the method of recruiting patients: we are talking about people with problems of very different clinical complexity, from abnormal heart rhythms to heart failure, therefore normally also people of very different ages and therefore also consequently with very different appearances. I have many doubts about recruiting via social media: why this decision? It wasn't explained enough.
--

	The same authors state that “The use of an online research platform for participant recruitment will disadvantage those who are less comfortable using online resources. This may include those living in deprived areas without access to high-tech quality ... and clinical populations with comorbidities that impede the use of certain technologies, such as visual, auditory, physical, or learning impairments.” But I would also add the majority of people with chronic heart disease such as heart failure, who would need these supports, but who find it more difficult to use them due to their age.
--	---

VERSION 1 – AUTHOR RESPONSE

Reviewer: 1

Comments to the Author:

The study protocol is very clearly written with references capturing international research underpinning the protocol.

We thank the reviewer for taking the time to read and provide feedback on our paper. We have addressed your concerns below.

1. The missing element in the text of the protocol (objectives, methods, study population) seems to be lack of recognition of digital exclusion e.g., the accessibility/availability of the focus group patients having the right technology/digital skills to participate in an online focus group via zoom; and usage of digital tools relating to heart disease clinical support.

Many thanks for raising this important point – we agree that accessibility is a key issue in this proposed work. Accordingly, we highlighted this as a limitation in the protocol (page 2):

“The use of an online research platform for participant recruitment will disadvantage certain demographic and clinical groups who are less comfortable using online resources. “

This protocol forms a small part of a much bigger research project, which has a theme dedicated entirely to issues surrounding accessibility and inclusivity. This point will also be considered within empirical papers resulting from this protocol.

Reviewer: 2

Comments to the Author:

This study is very interesting, and the goals are well designed. In fact, it is essential to have qualitative feedback from patients and clinicians on the use of technology at various levels and on their propensity.

Thank you for taking the time to read and provide such constructive feedback on our protocol. We have addressed the concerns you raised below.

1. The date of approval by the Ethics Committee is not disclosed. In addition, the protocol should indicate whether the study is already ongoing or planned, thus reporting specific dates.

Please specify if COREQ reporting guidelines were followed

Many thanks for highlighting this. We have clarified the status of the project at the timepoint of first submission to BMJ Open by adding the date of ethical approval and recruitment to the manuscript

(page 7 & 8). We have also emphasised the intention to write the resulting empirical paper according to COREQ guidelines (page 8).

“Recruitment and data collection began after ethical approval on 14/11/2022 and will continue until the end of March 2023.”

“This study was reviewed and approved by the Sciences & Technology Cross-School Research Ethics Council at the University of Sussex (reference ER/FM409/1) on 14/11/2022. We intend to write the resulting paper according to the Consolidated Criteria for Reporting Qualitative research (COREQ) guidelines (41).”

2. I have a strong doubt about the method of recruiting patients: we are talking about people with problems of very different clinical complexity, from abnormal heart rhythms to heart failure, therefore normally also people of very different ages and therefore also consequently with very different appearances.

We thank the reviewer for highlighting this point. We deliberately chose to recruit participants representing a diverse range of diagnoses and cardiological experiences. The aim of the overall project is to design a digital health tool which is diagnosis-agnostic and can accurately distinguish between subtle clinical characteristics. With this in mind, recruiting from a wide-pool of diagnostic, demographic and clinical presentations is critical to understand and inform our later developments.

Also, we do not feel it is necessary to discriminate between participants based on their diagnosis because our focus is on the healthcare experience prior to and during diagnosis which is unlikely to differ hugely between heart diseases as the individuals' symptoms have not yet been adequately investigated. We feel this allows us to capture a more holistic perspective of cardiovascular healthcare experiences than if we were to focus purely on one group of patients.

3. I have many doubts about recruiting via social media: why this decision? It wasn't explained enough.

We agree with the reviewer's concern, which is why we chose not to exclusively recruit via social media but to include it as one of three recruitment avenues that we implemented. We also reached out to existing clinical cohorts from previous research studies and used a targeted research recruitment platform (Prolific) which meant that we could recruit a cross-national cohort of both clinicians and lived experience groups (See page 7). Meanwhile, access to a community of GPs across the country via official Facebook groups increases our outreach and allows for a more diverse cohort to be included in our study.

“As we are interested in hearing from participants across a range of disease durations (i.e., those with very early symptoms and those who are on established management plans), we plan to recruit via: social media platforms such as LinkedIn, Twitter and Instagram; Prolific; and existing cohorts of people from the investigators' previous research studies who have consented to be contacted for research purposes. Clinicians will be recruited using purposive sampling via personal and professional connections, and social media platforms such as LinkedIn, Twitter and Instagram. “

We acknowledge that the use of social media to recruit participants may incur a certain bias in the recruited participant characteristics and intend to consider it as a point of discussion in the empirical papers that will come from this study.

4. The same authors state that “The use of an online research platform for participant recruitment will disadvantage those who are less comfortable using online resources. This

may include those living in deprived areas without access to high-tech quality ... and clinical populations with comorbidities that impede the use of certain technologies, such as visual, auditory, physical, or learning impairments."

But I would also add the majority of people with chronic heart disease such as heart failure, who would need these supports, but who find it more difficult to use them due to their age.

Thank you for making this point. To adhere to the editors' requirements, this statement has been substantially shortened in the revised submission (see page 2), but we will consider this point in depth in the write up of the resulting empirical paper.

"The use of an online research platform for participant recruitment will disadvantage certain demographic and clinical groups who are less comfortable using online resources. "

Editor Comments to Author:

We thank the Editor for taking the time to read and provide constructive feedback on our paper. We have addressed your concerns below.

1. Please update the 'Methods and Analysis' section of the abstract to provide more information about the study setting.

We have added a statement in the abstract to show that this will be a decentralised study that will recruit a nationwide cohort (page 1).

"Methods and Analysis A series of qualitative studies will be carried out with the aim of developing a stakeholder-led perspective on the implementation of digital technologies to improve holistic diagnosis of heart disease. This will be a decentralised study with all data collection being carried out online with a nationwide cohort. Focus groups will be carried out with 20 people with lived experience of heart disease, and interviews will be carried out with 10 clinicians with experience of diagnosing heart diseases. The data will be analysed using an inductive thematic analysis approach."

2. Please update the 'Ethics and Dissemination' section of the abstract to include the reference number for the ethics approval and brief mention of participant informed consent requirements.

We have added the reference number for ethics approval and specifics relating to the informed consent requirements in the "ethics and dissemination" section of the abstract (page 2)

"Ethics and Dissemination

This study received ethical approval from the Sciences and Technology Cross Research Council at the University of Sussex (reference ER/FM409/1). Participants will be required to provide informed consent via a Qualtrics survey before being accepted into the online interview or focus group. The findings will be disseminated through conference presentations, peer-reviewed publications and to the study participants."

3. Please revise the third bullet point in the 'Strengths and limitations of this study' section so that it consists of a single sentence.

As requested, we have revised the third bullet point to be more concise (see page 2)

“The use of an online research platform for participant recruitment will disadvantage certain demographic and clinical groups who are less comfortable using online resources. ”

4. In the main text ‘Study Population’ section, please provide an explanation of how the sample sizes were determined. For example, was this based on assumed number needed to reach thematic saturation? Or was the sample size based on logistical or feasibility considerations? Will saturation be assessed?

A brief statement has been added to explain how the projected sample size were determined (see page 6).

“Based on the available time for data collection against the wider project deadlines, we plan to conduct four focus groups with individuals with lived experience of heart disease. We aim to recruit 5-6 individuals per focus group to ensure there is enough time for each participant to share their views and experiences.”

“Based on the research team's previous experience of conducting qualitative research with clinicians, a pragmatic decision has been made to aim for a target of 10 clinicians who have had any experience (past or present) of diagnosing people with heart disease to be enrolled into interviews.”

Furthermore, in line with recent developments in the literature (Braun & Clarke, 2019), saturation will not be assessed as ideas of thematic saturation are not consistent with the values and assumptions of reflexive thematic analysis.

VERSION 2 – REVIEW

REVIEWER	Palmira Bernocchi ICS Maugeri SpA SB, Telemedicine Service
REVIEW RETURNED	17-May-2023
GENERAL COMMENTS	The authors have extensively and exhaustively responded to requests. I have no further requests.